# The Contribution of Actinobacteria to the Degradation of Chlorinated Compounds: Variations in the Activity of Key Degradation Enzymes

**DOI:** 10.3390/microorganisms11010141

**Published:** 2023-01-05

**Authors:** Elena V. Emelyanova, Sudarsu V. Ramanaiah, Nataliya V. Prisyazhnaya, Ekaterina S. Shumkova, Elena G. Plotnikova, Yonghong Wu, Inna P. Solyanikova

**Affiliations:** 1G.K. Skryabin Institute of Biochemistry and Physiology of Microorganisms, Pushchino Center for Biological Research of the Russian Academy of Sciences, Prosp. Nauki 5, 142290 Pushchino, Russia; 2Food and Biotechnology Research Lab., South Ural State University (SUSU), 76, Lenin Prospekt, 454080 Chelyabinsk, Russia; 3FBGUN Institute of Molecular Biology V.A. Engelhardt RAS, 119991 Moscow, Russia; 4Institute of Ecology and Genetics of Microorganisms, Perm Federal Research Center, Ural Branch of the Russian Academy of Sciences, 614081 Perm, Russia; 5Zigui Ecological Station for Three Gorges Dam Project, State Key Laboratory of Soil and Sustainable Agriculture, Institute of Soil Science, Chinese Academy of Sciences, 71 East Beijing Road, Nanjing 210008, China; 6Regional Microbiological Center, Institute of Pharmacy, Chemistry and Biology, Belgorod National Research University, 308015 Belgorod, Russia

**Keywords:** actinobacteria, pollutants, degradation, enzymes, specificity, mega-plasmid

## Abstract

Bacteria make a huge contribution to the purification of the environment from toxic stable pollutants of anthropogenic and natural origin due to the diversity of their enzyme systems. For example, the ability to decompose 3-chlorobenzoate (3CBA) by the four representative genera of Actinobacteria, such as *Rhodococcus*, *Gordonia*, *Microbacterium*, and *Arthrobacter*, was studied. In most cases, the formation of 4-chlorocatechol as the only key intermediate during the decomposition of 3CBA was observed. However, *Rhodococcus opacus* strain 1CP was an exception, whose cells decomposed 3CBA via both 3-chloro- and 4-chlorocatechol. The enzyme 3-Chlorobenzoate 1,2-dioxygenase (3CBDO) induced during the growth of these bacteria in the presence of 3CBA differed significantly in substrate specificity from the benzoate dioxygenases induced upon growth in the presence of benzoate. The *R. opacus* 6a strain was found to contain genes encoding chlorocatechol 1,2-dioxygenase, chloromuconate cycloisomerase, and dienelactone hydrolase, whose nucleotide sequence was 100% consistent with the sequences of the corresponding genes encoding the enzymes of the modified 4-chlorocatechol *ortho*-cleavage pathway of the strain *R. opacus* 1CP. However, the gene encoding chloromuconolactone dehalogenase (*clcF*) was not found in the representatives of the actinomycete genera, including *Gordonia* and *Arthrobacter*. A linear mega-plasmid carrying 3-chlorocatechol degradation genes remained stable after maintaining the *R. opacus* 1CP strain on an agar-rich medium for 25 years. In general, a similar plasmid was absent in actinobacteria of other genera, as well as in closely related species of *R. opacus* 6a.

## 1. Introduction

Almost all natural (both biogenic and abiogenic) origin chlorinated aromatic compounds and those that enter the environment because of human activities have a negative impact on living organisms [1,2]. Bacteria possess a lot of biodegradation enzymes to degrade various organic compounds [3,4]. To date, the transfer of the ability to degrade resistant xenobiotics among microorganisms occurring through the horizontal transfer of biodegradation plasmids is considered proven. Such exchange of information can occur even among microorganisms belonging to different taxonomic groups [5,6].

The history of the study of bacterial degradation pathways of monoaromatic compounds dates back several decades [7]. It was shown that the introduction of chlorinated substituents into the aromatic ring increases the resistance of this molecule to microbial attack. In general, the more substituents the ring contains, the more resistant it is to decomposition [8]. The ability of bacteria to successfully attack the chlorocatechol ring was first studied in *Pseudomonas* sp. B13 and *Alcaligenes* sp. strain A7-2 [9]. The conversion of 3-chlorobenzoate (3CBA) into intermediates of the tricarboxylic acid cycle requires the presence of a special set of enzymes, including chlorocatechol 1,2-dioxygenase (CCat 1,2-DO) and chloromuconate cycloisomerase (CMCI). The substrate specificities of these enzymes differed significantly from the specificities of similar enzymes of the studied strain, mediating unsubstituted benzoate decomposition with the formation of catechol as a key intermediate [10]. Two more enzymes (dienelactone hydrolase and maleylacetate reductase) were absent in the catechol conversion pathway and played an important role in the transformation of intermediates during the cleavage of chlorocatechols [11,12]. Thus, the past and present studies clearly indicated that the presence of enzymes, such as CCat 1,2-DO, CMCI, and DLH, with new substrate specificity, was needed for the successful conversion of mono- and dichlorinated aromatic compounds when chlorocatechols are formed as key intermediates.

Chlorobenzoates as herbicides are widely used in agriculture. The compound 3CBA is also applied in industry as a raw material for the fabrication of dyes, pharmaceutical preparations, and fungicides and as a preserving agent for adhesives and dyes. This compound can accumulate in the environment as a result of the co-metabolism of polychlorobiphenyls and chlorotoluenes and water chlorination. Therefore, 3CBA is detected in industrial wastewater, rivers, and underground water [10]. The compound 3CBA turned out to be one of the compounds that many researchers used as a “model” to study the ability of microorganisms to decompose chloroaromatic compounds and to identify the features of the enzymes involved in this process [10]. Significant progress has been achieved in this area in a relatively short time. However, the reaction of the initial conversion of not only 3CBA but also a number of other monoaromatic compounds by bacteria of different groups still remains poorly studied. It is known that if dehalogenation does not occur at the first stage, then the decomposition of 2-chlorobenzoate (2CBA) and 2-chlorophenol (2CP) or 4-chlorobenzoate and 4-chlorophenol (4CP) is accompanied by the formation of 3-chlorocatechol (3CCat) or 4-chlorocatechol (4CCat), respectively. The transformation of 3-chlorophenol (3CP), 3-chloroaniline (3CA), and 3CBA remain the most unclear since, in this case, both 3CCat and 4CCat can be formed. The ratio of the resulting products will depend on the substrate specificity of the enzymes of the initial attack, benzoate dioxygenase or phenol hydroxylase [13].

Substrate specificity of benzoate dioxygenases from several Gram-negative bacteria, *Pseudomonas* sp. B13 and *Pseudomonas putida* pAC 27 have been studied [7]. The study [14] showed that the formation of 3CCat and 4CCat occurs in the 66: 33 ratio. The process of transformation of a number of mono-chloroaromatic compounds bearing a substituent in the *meta*-position under the action of *Rhodococcus* has been studied. The results allowed the conclusion that 4CCat is the only product formed due to the specificity of initial attack enzymes. Thus, based on the previously presented works, it can be assumed that the further conversion of 3CBA and 3CP by rhodococci occurs with the formation of only 4CCat, while Gram-negative bacteria can produce both 3CCat and 4CCat. However, we have previously shown that the degradation of 3CBA by the *Rhodococcus opacus* 1CP strain proceeds under the action of the enzymes of a newly modified *ortho* pathway with 3CCat formed as a key intermediate [15]. The purpose of this work was to identify the type of chlorocatechol dioxygenase induced during the decomposition of 3-chlorobenzoate by Gram-positive bacteria. In fact, that the ability of Gram-positive bacteria to degrade 3CBA has not yet been sufficiently studied, and the information on the properties of enzymes involved in the decomposition of 3-chlorocatechol in *Rhodococcus* is extremely limited, we chose this compound for further study of the processes of microbial conversion of chlorobenzoates. In our opinion, the clarification of the situation with the set of enzymes possessed by *Rhodococcus* is a clear example of the role of microorganisms in cleaning and restoring the health of the environment.

## 2. Material and Methods

### 2.1. Bacterial Culture and Cultivation Conditions

In this work, bacterial strains isolated from soil contaminated with (halogen) aromatic compounds (OAO Halogen, Perm, Russia); *Rhodococcus* sp. P1, P20 [16], *Rhodococcus ruber* P25 (=IEGM896) [17], *Rhodococcus* sp. G10 [18] and previously described *R. opacus* 1CP and *Rhodococcus rhodochrous* 89 [19,20] were used. The *Rhodococcus opacus* 6a strain was isolated using 4-chlorophenol as a carbon source by the method of enrichment cultures from the soil taken from the territory of the gas station. Cells were grown in mineral medium G with the following composition (g/L): Na_2_HPO_4_, 0.7; KH_2_PO_4_, 0.5; NH_4_NO_3_, 0.75; MgSO_4_ × 7H_2_O, 0.2; MnSO_4_, 0.001; FeSO_4_, 0.02 [19], containing 3-chlorobenzoate sodium salt as the sole source of carbon and energy.

To obtain cell biomass, the strains were grown in the presence of 3-chlorobenzoate 3CBA as a carbon source in Erlenmeyer flasks containing 200 mL of the medium. The compound 3CBA was added at a concentration of 200 mg/L. The growth of cultures was monitored by spectrophotometric determination of the residual amount of the substrate, taking spectra from 220 nm to 340 nm after cell sedimentation on an Eppendorf 5414 centrifuge (Hamburg, Germany). The pH of the growth medium was maintained at a level of 7.0–7.2 by periodic additions of 0.1 M NaOH. The final optical density at 545 nm was 1.6–1.8. Cells were pelleted by centrifugation (7000 rpm, 10 min), washed twice with 50 mM Tris-HCl buffer, and stored at −20°C. To determine the activity of benzoate dioxygenase (see Section 2.8), the cells were not frozen but used immediately after washing with buffer.

### 2.2. PCR

Amplification of the genes encoding chloromuconolactone (CMLI) was performed using the primers clcF_fwgcttcgacatatgttgtacctagttc and clcF_revatggatcctcagtctttgccgac [21] under the following reaction conditions: initial denaturation, 5 min at 95 °C, then 24 cycles: 30 s—94 °C, 30 s—40 °C, 45 s—72 °C; 1 cycle: 10 min—72 °C. The reaction mixture contained 1.5 mM MgCl_2_, 0.5 µM of each primer, 200 µM dNTP, and 2.5 Unit Tag polymerase (Silex, Moscow, Russia). The primers allowed amplification of the complete *clcF* gene, which was 276 bp in *R. opacus* 1CP (GenBank AJ439407). The reaction products were separated by agarose gel electrophoresis (1%) at a voltage of 10 V/cm, stained with ethidium bromide solution (5 μg/mL), and photographed under UV light using the GelDocTMXR gel-documentation system (Bio-Rad Laboratories, Hercules, CA, USA).

### 2.3. Sequencing and Analysis of Nucleotide Sequences

Nucleotide sequences were determined using the Big Dye Terminator Ready Reaction Kit v 3.1 (Applied Biosystems, Foster City, CA, USA) on a Genetic Analyzer 3500XL automated sequencer (Applied Biosystems, USA) according to the manufacturer’s recommendations. The search for homologous sequences was performed using the GenBank (http:/www.ncbi.nlm.nih.gov; 8 October 2022) and EzTaxon (http://www.eztaxon.org; 8 October 2022) databases. The percentage of similarity of genes with 16S rRNA with homologous genes of type strains was calculated using the online resources of the EzTaxon server. (http://www.eztaxon.org; 8 October 2022). The search for homologues of the clcF genes and their preliminary analysis was performed using the BLAST programs (http://www.ncbi.nlm.nih.gov; 8 October 2022).

### 2.4. Identification of Plasmid DNA

The presence of plasmid DNA was detected by pulse electrophoresis using a CHEF DR II device (Bio-Rad Laboratories, Hercules, CA, USA). The strain was grown in 10 mL of bacterial survival ratio (BSR) without adding NaCl or in 10 mL of magnetospirillum (MSR) containing 10 mg/mL of NaCl and one of the hydrocarbons (o-FA, BA, biphenyl (1g/L)), to OD600 = 1.0. Cells were pelleted by centrifugation (9660× *g*, 3 min) and washed twice in TE buffer (10 mM Tris/HCl, pH 7.6; 1 mM EDTA, pH 8.0). Agarose blocks were prepared according to the manufacturer’s recommendations (Bio-Rad Laboratories, Hercules, CA, USA). The blocks were treated with lysozyme (1 mg/mL) at 37 °C for 5–16 h, with proteinase K (1 mg/mL) at 50 °C for 12–18 h, with nuclease S1 (5 units per agarose block) at 37 °C, 3.5 h. Sample electrophoresis was performed in 1% agarose gel (Pulsed Field Certified Agarose, Bio-Rad Laboratories, Hercules, CA, USA) in 0.5 TBE buffer (108 g Tris, 55 g boric acid, 40 mL 0.5 M EDTA, up to 1 L H_2_O) at 14 °C, 6 V/cm, pulse time from 60 s to 120 s, for 24 h. The gel was stained with ethidium bromide (0.5 mg/L, 10 min) and photographed under ultraviolet light using a gel documentation system (Bio-Rad Laboratories, Hercules, CA, USA). The size of extrachromosomal DNA was estimated in comparison with the electrophoretic mobility of the DNA Size Markers—Yeast Chromosomal molecular weight marker (Bio-Rad Laboratories, Hercules, CA, USA).

### 2.5. Preparation of Cell-Free Extract

To prepare a cell-free extract, the biomass was destroyed by extrusion disintegration on a Hughes-type press with an operating pressure of 3200 kg/cm^2^. After disintegration, the biomass was incubated with DNase for 15 min at room temperature and centrifuged at 15,000× *g* for 40 min at 5 °C. The pellet was resuspended in 50 mM Tris-HCl buffer (pH 7.2) and centrifuged under the same conditions. The resulting supernatants were pooled and centrifuged again for enzyme tests and enzyme purification.

### 2.6. Determination of Enzyme Activity and Protein Amount

Enzyme activity was determined spectrophotometrically on a Shimadzu UV-160 instrument (Kyoto, Japan) in quartz cuvettes with an optical path length of 1 cm at 25 °C. Chlorocatechol 1,2-dioxygenase activity was determined using a modified method by Hayaishi et al. [22]. The reaction mixture contained 0.25 mM catechol or substituted catechols, 1.3 mM EDTA, and an enzyme in 50 mM Tris-HCl buffer (pH 7.2). The reaction was started by adding the enzyme. Enzyme activity was calculated from the rate of formation of the product, cis,cis-muconic acid, or substituted muconates, at a wavelength of 260 nm. Chloromuconate cycloisomerase activity was determined using the method described by Schmidt and Knackmuss [23]. The reaction mixture contained 50 mM Tris-HCl buffer (pH 7.2), 2 mM MgCl_2_, 0.1 mM cis,cis-muconic acid, and enzyme. The reaction was started by adding the enzyme. The reaction rate was determined by the disappearance of the substrate at a wavelength of 260 nm. The activity of dienelactone hydrolase (EC 3.1.1.45) was determined according to the method described by Moiseeva et al. [24]. The reaction mixture contained: 50 mM Tris/HCl, pH 7.2, 0.1 mM substrate, and enzyme. The reaction was started by adding the enzyme. DLH activity was calculated from the rate of loss of cis-dienelactone at 280 nm. When calculating the activities, the molar extinction coefficients determined by Dorn and Knackmuss [25] were used: 16,800 M^−1^cm^−1^ for catechol; 17,100 M^−1^cm^−1^ for 3-chlorocatechol; 12,400 M^−1^cm^−1^ for 4-chlorocatechol; 12,000 M^−1^cm^−1^ for 3,5-dichlorocatechol; 18,000 M^−1^cm^−1^ for 3-methylcatechol; 13,900 M^−1^cm^−1^ for 4-methylcatechol, 17,000 M^−1^cm^−1^ for cis-dienelactone.

A unit of enzymatic activity was defined as the amount of enzyme catalyzing the conversion of 1 µmol of substrate or the formation of 1 µmol of product per minute. Relative activity was calculated as 100% activity with an unsubstituted or better substrate.

The protein concentration was determined according to the modified method of Bradford [26], using bovine serum albumin as a standard.

### 2.7. Purification of Enzymes from the Biomass of R. opacus 6a, R.opacus 1CP, R.ruber P25, Microbacterium Foliorum B51 Strains Grown with 3-CBA as Substrate

The cell-free extract was applied to a Q-Sepharose column (26 × 20, volume 80 mL) equilibrated with 50 mM Tris-HCl buffer, pH 7.2 (buffer A). The column was washed with one volume of the same buffer, after which the protein was eluted with a linear gradient (0–0.5 M) of NaCl in 300 mL of buffer A at a rate of 1 mL/min. Fractions with a volume of 5 mL were collected. The active fractions were combined, and an equal volume of 1.6 M ammonium sulfate was added to the obtained protein solution. After centrifugation at 16,000× *g* for 30 min, the protein solution was applied to a Phenyl-Sepharose column (26 × 40, volume 133 mL) equilibrated with 50 mM Tris-HCl buffer, pH 7.2 (buffer B) with 0.8 M ammonium sulfate. The solution was eluted with a linear (NH_4_)_2_SO_4_ gradient (0.8–0 M) in 2000 mL of buffer B at a rate of 2 mL/min. Fractions of 10 mL with maximum activity were pooled and concentrated to 2 mL. The preparation was applied to a Superdex 200 column (16 × 70, volume 120 mL) equilibrated with buffer B with 0.1 M NaCl and eluted with the same solution at a rate of 1 mL/min. The active fractions, 1.5 mL each, were combined, desalted, and concentrated in a 50 mL Amicon cell (Millipore Corporation, Burlington, MA, USA) using a cellulose ultrafiltration membrane with a pore diameter of 30,000 μm to 2 mL (Sigma, St. Louis, MO, USA). Catechol was used as a substrate during purification.

### 2.8. Polarographic Measurement of Cell Respiration and BDO Activity in Bacterial Cells

To determine 3-chlorobenzoate 1,2-dioxygenases (3CBDO) activity, biomass grown in a liquid mineral medium containing 200 mg/L of the sodium salt of 3-chlorobenzoate was prepared and immediately analyzed. Measurements were carried out according to the method previously described by Emelyanova and Solyanikova [27]. In this case, the signal reflected the rate of change in respiration in the presence of a substrate of 3CBDO (the signal was expressed as μg O_2_ (L s^−1^)) and characterized the activity of the enzyme. To determine the activity of monooxygenase, a biosensor technique was applied. A laboratory model of a membrane microbial sensor with an oxygen electrode as a transducer was formed based on immobilized actinobacterial cells. The change in respiration of microbial cells in response to enzyme substrate was expressed as pA s^−1^, which was proportional to the enzyme activity [28].

### 2.9. Matrix Assisted Laser Desorption Ionization-Time of Flight Mass Spectrometry (MALDI-TOF MS) Analysis

Preparation of cells for analysis: whole cells were used for analysis. Cultures were grown on mineral agar medium with 3CBA as a carbon source. Cells (~5–10 μg) were transferred with a thin, sterile spatula to a plastic tube with 50 μL of freshly prepared 50% aqueous acetonitrile (Sigma-Aldrich) containing 2.5% trifluoroacetic acid and mixed thoroughly. The resulting suspension (0.8 μL) was applied to a steel target plate and mixed with an equal volume of the matrix solution (α-cyano-4-hydroxycinnamic acid in 50% aqueous acetonitrile containing 0.1% trifluoroacetic acid) and dried on air at room temperature. The spectra were recorded using an Autoflex Speed mass spectrometer (Bruker Daltonics) according to the manufacturer’s recommendations. The instrument was calibrated using Protein 1 Calibration Standard (Bruker Daltonics); the resolution of the spectra was ±2 Da (200 ppm). The resulting spectra of each preparation of strains were obtained by summing the spectra recorded at 5–10 points of the analyzed preparations at 500 laser strokes. Mass spectra were processed using the Flex analysis 3.3 software package (Bruker Daltonics).

## 3. Results

### 3.1. The Activity of Actinobacteria with Substituted Benzoate and Phenols

As shown earlier, several actinobacteria used 3CBA as the best growth substrate [29]. For example, utilization of 500 mg/L 3CBA by cultures of *Microbacterium foliorum* B51, *R. wratislaviensis* P1, and G10 occurred in less than 12 h. *Rhodococcus opacus* 6a cells degraded to 500 mg/L 3CBA in less than 24 h. Although the other cultures were not as efficient as listed above, a culture of *Rhodococcus rhodochrous* 89 was also able to grow in the liquid medium in the presence of 3CBA. The inducible enzyme 3CBDO was initiated by the degradation of 3CBA and catalyzed in an oxygen-dependent reaction. This enzyme has a complex structure, and its activity could not be determined in cell-free extracts as the enzyme was damaged when bacterial cells were destroyed. Therefore, enzyme activity was determined in whole cells by determining the change in the rate of oxygen consumption by bacterial cells in the presence of the substrate of the enzyme. If the substance was not a substrate for the enzyme, then cell respiration did not change in the presence of this substance. Actinobacterial cell responses to benzoate, mono- and di-substituted benzoates, phenol, and mono-substituted phenols were determined. Data on the substrate specificity of 3CBDO of actinobacterial cells induced by 3CBA (growth of cells in the presence of 3CBA) were presented in Table 1. The response of cells to substrates was given in % of the response to 3CBA6. The response to 3CBA was taken as 100%.

The obtained results indicated that 3CBDO was not of narrow substrate specificity. The high values of response to most substrates besides 3CBA have been obtained for *R. wratislaviensis* G10. Moreover, the response of *G. polyisoprenivorans* 135 cells to chlorinated phenols was compared to their response to 3CBA.

The results in Table 1 showed that 3CBDO from *R. opacus* 1CP cells was characterized by broad substrate specificity when compared to BDO from the same culture. For *R. opacus* 1CP, the magnitudes of 3CBDO activity with BA (30%) and 3CBA (100%) indicated that the affinity of 3CBDO for 3CBA was higher than that for BA. Using the biosensor technique, we have previously shown that the magnitude of the constant characterizing the affinity of 3CBA for 3CBDO (484 μM) was three times less than the magnitude of the constant for the 3CBA-BDO pair (1400 μM) [30]. For *R. opacus* 1CP, the response was obtained for all three chlorinated benzoates (2, 3, and 4CBA). Moreover, the magnitude of the response to 2CBA was comparable to the magnitude of the response to 3CBA, but the response to 4CBA was significantly lower. While *R. rhodochrous* 89 did not respond to 2CBA, the response to 4CBA by 1CP cells was significantly lower than 100%. For *R. wratislaviensis* G10, a response was recorded to 2CBA, and it was several times higher than the response to 3CBA. Perhaps this could be explained by the features of 3CBDO previously found for 3CBDO of 1CP cells [27]. Studying the competition of 3CBA and its analogs 2CBA and 4CBA for 3CBDO, it was concluded that these substrates bound to the enzyme in different ways. It was assumed that 2CBA bound to the active site of the enzyme, competing with 3CBA for 3CBDO. Unlike 2CBA, 4CBA is most likely bound to the regulatory sites of 3CBDO. Therefore, 4CBA could not be metabolized by 3CBDO of *R. opacus* 1CP cells.

3CBDO of *R. opacus* 1CP was active not only in the presence of substituted benzoates but also in the presence of chlorinated phenols (Table 1). This is due to the peculiarities of the gene apparatus of the culture, in which the encoded enzymes mediated the degradation of a wide range of pollutants. Based on immobilized cells of *R. opacus* 1CP actinobacteria, the response to acetone for induced cells was recorded using a biosensor technique. Significant activation of respiration in the presence of acetone was observed for actinobacterial cells induced by this substrate (Figure 1a). Moreover, for the enzyme system that induced the metabolism of acetone, positive kinetic cooperativity for the substrate has been found, as for the BDO of 1CP cells. This was evidenced by the non-linearity of the plot of the double-reciprocal dependency; the resulting curve was convex (Figure 1b)

### 3.2. Enzymes of Strains R. opacus 6a, R. opacus 1CP, R. ruber P25, M. foliorum B51 Grown on 3-CBA

In cell-free extracts of the studied strains, the activity of enzymes of the modified *ortho*-pathway, which were involved in the decomposition of chloroaromatic compounds, was determined. In cell-free extracts of 3-CBA grown *R. opacus* 1CP, *R. ruber* P25, and *M. foliorum* B51 strains, both CCat 1, 2-DO, and DLH activity was shown. The most unobvious picture was with CMCI. Only the strain *R.opacus* strain 1CP showed low activity with muconate (0.003 µmol/min/mg protein). Activity with 2-methylmuconate was found only in the strain *R.ruber* P25. However, activity with 3-methylmuconate was detected for all strains, except for *M. foliorum* B51, while activity with 2-chloromuconate was observed for *R. opacus* 6a, *R. opacus* 1CP, and *R. ruber* P25 (Table 2).

The substrate specificity of Cat 1,2-DO in a cell-free extract of the *R. opacus* 6a strain was investigated. Analysis of the study showed the presence of catechol 1,2-dioxygenases activity with 3CCat, equal to the activity with Cat, against the relatively low activity with 4CCat (25%) (Table 3). However, since in all cases, the activity of dioxygenase with 4-chlorocatechol was relatively high, all enzymes tested could be tentatively designated as chlorocatechol 1, 2-dioxygenase.

It was found for 3CBA-grown *R. ruber* P25 strain that the relative activity of chlorocatechol 1, 2-DO with 3MCat (125%) was slightly higher than that with Cat, and the activity with 4MCat (162%) was significantly higher. Whereas, when *R. ruber* P25 was grown in the presence of *para*-toluate, the activity of Cat 1,2-DO with 3MCat (195%) was higher than with 4MCat (117%) [31]. For chlorocatechol 1, 2-DO induced by 3CBA, the presence of relatively high activity with 4CCat indicated that the P25 strain could have several dioxygenases differing in substrate specificity. In the case of chlorocatechol 1, 2-DO, isolated from 3CBA-grown *R.opacus* 1CP strain, the relative activity with 3MCat and 4MCat was 120 and 197, respectively. Whereas for 2CP-grown *R. opacus* 1CP, the activity was 283 and 270, respectively [24]. For *M. foliorum* B51 strain, the activity of chlorocatechol 1,2-DO with 4MCat was quite high when compared to 3MCat (Table 3).

Using the example of chlorocatechol 1,2-DO from Gram-positive bacteria grown in a medium containing the same substrate (in this case, 3CBA), the study showed that the induced enzymes could be significantly different in substrate specificity. However, these differences could result from both the induction of enzymes with different properties and the presence of several CCat 1,2-DO isoforms of enzyme in the cell-free extract. To find out what were the reasons for the unusual properties of some dioxygenases, we tried to determine the number of isoforms of each enzyme.

As a result of the purification of enzymes from the biomass of 3CBA-grown *R. opacus* 1CP strain, dioxygenase activity was detected in the form of a single peak at each stage of purification. Previous research has shown that CCat 1,2-DO was induced by only one dioxygenase isoform when this strain was grown in the presence of 2-CP and 4-CP [24]. The activity of CCat 1,2-DO was detected with 3, 5-DCCat (233%) (Table 4), which indicated the possible involvement of an enzyme with an unusual substrate specificity in this bacterium after a resting state in the process of 3CBA destruction.

*R. ruber* P25 strain had one peak of dioxygenase activity. The activity of CCat 1,2-DO from *R. ruber* P25 was 100%, 225%, 262%, and 87% with Cat, 3MCat, 4MCat, and 4CCat, respectively. The obtained data indicated that, in terms of substrate specificity, this dioxygenase differs from the dioxygenase induced during the growth of the P25 strain on 4-methylbenzoate when activity was higher with 4CCat and 3,6DCCat.

For the *M. foliorum* strain B51 after ion-exchange chromatography, two peaks of dioxygenase activity were detected. The analysis of the study showed that the enzyme from the first peak belongs to the catechol 1,2-dioxygenase of the ordinary *ortho*-cleavage pathway since it had no activity with chlorocatechols. In contrast, the enzyme from the second peak could be attributed to chlorocatechol dioxygenases with rather high activity toward 4CCat (77%). This enzyme was also active with 3,5-DCCat (9.8%). In terms of substrate specificity, this enzyme was closely related to the analogous enzyme of 4CP-grown *R.opacus* 1CP. This study showed the presence of enzymes with different substrate specificities. The change in the substrate specificity of dioxygenases during purification indicated the presence of several isoenzymes in the 1CP strain. This feature allowed strain to adapt to new growth substrates. One peak of dioxygenase activity was detected for studied 3CBA-grown rhodococci. On the contrary, for *R. opacus* 1CP grown with any of the chlorinated substrates, only CCat 1,2-DO was induced; on the medium containing unsubstituted benzoate and phenol, only Cat 1,2-DO was induced. *M. foliorum* B51, such as *Pseudomonas*, differed from *Rhodococcus* in the presence of two peaks of dioxygenase activity.

### 3.3. clcF Gene in Actinobacteria

Several strains capable of growing with chlorobenzoic acids were checked for the presence of the *clcF* gene encoding 5-chloromuconolactone dehalogenase (5CMLD) and the 4CCat degradation enzymes. The enzyme 5CMLD caused the ability of the strain *R. opacus* 1CP to use 2-chlorophenol as the sole source of growth, decomposing 3-CCat, formed from 2-CP, via a new modified *ortho*-pathway. The presence of the *clcF* gene in the *R. opacus* strain 1CP was shown previously [21]. This gene was discovered more than 10 years ago in a culture adapted to 2-CP after growth with 4-CP. An interesting fact was that the study was able to identify the gene in cells that were stored in the laboratory for more than 25 years on a rich medium (Figure 2). The obtained data were in some contradiction with the existing opinion that the ability of bacteria to degrade pollutants, mediated by the presence of biodegradation plasmids, was quickly lost in the absence of exposure to a toxicant as a result of the elimination of the corresponding plasmids. In this case, it was known that the genes of the new modified *ortho* pathway in the *R. opacus* 1CP strain were located on the linear plasmid [32]. The preservation of at least one gene of this pathway under non-selective conditions may indicate that the strain stably retained this plasmid when grown on a rich medium. The assumption that recombination could occur during storage and that this gene was also presented in the chromosome seems unlikely possible. In general, it was not fundamental since, in this case, the fact of preserving the phenotype during storage on a rich medium was important. The safety of the linear plasmid is discussed below.

Checking the distribution of the gene encoding CMLD among aromatic decomposer strains showed that the exact copy of it was found in several aromatic decomposer strains. At the same time, bacteria of the species *Rhodococcus wratislaviensis*, in the genome of which this gene was found, were isolated from the sites contaminated with haloaromatic compounds in the territory of the Perm and were characterized by the ability to decompose chlorobiphenyls. As already known, the decomposition pathway of chlorobiphenyls did not imply the formation of 3-CCat as a key intermediate. In this case, it indirectly evidenced that the inheritance of this gene also occurred in the absence of selective pressure. However, the study had no evidence of whether there was no selective pressure on these strains under field conditions (soils from the territory of OAO Halogen). The pollution was various compounds, including haloaromatic ones, and there also could be 3CBA (although now it is mainly organofluorine and organobromo). The company was founded in 1942 (http://b2bpoisk.ru/%D0%BA%D0%BE%D0%BC%D0%BF%D0%B0%D0%BD%D0%B8%D1%8F/%D0%BE%D0%B0%D0%BE_%D0%B3%D0%B0%D0%BB%D0%BE%D0%B3%D0%B5%D0%BD; 21 February 2022). Currently, this enterprise produced more than 100 types of various chemical compounds, and in total officially 170 types of various products were produced. In general, it was not known what bacteria acquired earlier: 3-CBA destruction genes, including *clcF*, or genes of the upper biphenyl (CB) destruction pathway. Theoretically, the conditions (the composition of compounds in the soil) could probably change as the types of manufactured products changed or the decline in production after 1980. Thus, based on the obtained data, it could be concluded that the newly modified *ortho* pathway genes were widely distributed in nature and inherited stably even in the absence of selective pressure.

DNA from bacteria *G.polyisoprenivorans* 135, *R. rhodochrous* 89, and *A. agilis* Lush13 was used as a template to test for the presence of the gene encoding CML. The obtained data indicated that these strains did not have a gene identical to that found in bacteria of *R. wratislaviensis* species, including strains P20 and P1. Thus, the study showed that bacteria of the genus *Rhodococcus* had a wide range of enzymes involved in the decomposition of chloroaromatic compounds via modified pathways, and the similarity of these enzymes to each other was much higher than among similar enzymes of gram-negative bacteria (Figure 3).

### 3.4. MALDI-ToF Mass Spectrometric Analysis as a Method for Rapid Screening of Enzymes of Interest

In recent years, for the diagnosis of microorganisms, such an express method as MALDI-ToF mass spectrometry has been actively developed and used, especially in medical practice. We attempted to adapt this method for the rapid screening of strains for their ability to degrade target substrates and to determine the pathway by which they are degraded. An analysis for the presence of CMLD was used as a standard test. In this situation, this enzyme could serve as the basis for separating the flows along which the decomposition of chloroaromatic compounds in actinobacteria proceeds: via the path of *ortho*-cleavage of 3-CCat, 4-CCat or with the stage of initial dehalogenation. Decomposition of 2-chloroaromatic compounds may, in the first step, include either dehalogenation to form unsubstituted pyrocatechol or aromatic ring hydroxylation to form 3CCat. In the first case, the cleavage pathway of unsubstituted catechol did not imply the induction of CMLD; in the second case, CMLD was involved in the conversion of 3-CCat via a new modified *ortho*-cleavage pathway. The decomposition of 3-chloro-substituted substrates under the action of dioxygenases could be accompanied, depending on their specificity, by the formation of 3-CCat or 4-CCat. In the first case, it can be assumed that CMLD was involved; in the second case, CMLD was not involved in the conversion of 4-CCat. The decomposition of 4-chloroaromatic compounds could also occur both with initial dehalogenation and with the formation of 4-CCat. However, in both cases, CMLD did not participate in the transformations. In this work, the use of CMLD as a standard made it possible to overcome one significant drawback of the MALDI-ToF MS method. A significant limitation on the use of this method was exerted by the masses of the analyzed proteins. Unless any pretreatment of whole cells or protein solutions with lytic agents was expected, the use of MALDI was generally limited to the masses that could be recorded by this instrument. In the case of using CMLD, there was no need for any preliminary processing of the analyzed sample since the mass of the CMLD subunit was significantly smaller than the mass detection limit of this device.

A previous study showed that the analysis of both whole cells of the *R. opacus* 1CP strain grown on 3-CBA and CMLD isolated from them revealed a peak with a mass of 11,193 Da [33], which corresponded well to the derived mass of CMLD, which indicated the formation of 3-CCat during the growth of this culture with 3-CBA (Figure 4). Analysis of partially purified protein preparations of *R. ruber* P25 and *M. foliorum* B51 also produced corresponding mass peaks. However, analysis of whole cells of *R. wratislaviensis* G10, P20 strains grown on an agar mineral medium with 3-CBA as the only growth substrate by MALDI spectrometry did not reveal any peaks characteristic of the mono-, di-, or triply charged ClcF ion, which was good and consistent with the lack of visible activity in the cell-free extract (Figure 4).

The ability of bacteria to decompose stable pollutants was determined by the presence of appropriate enzymes in them. The genomes of most rhodococci were large and contained a significant amount of biodegradation genes. Thus, the genome of *Rhodococcus erythropolis* B7g isolated with toluene comprises 7,175,690 bp [34]. The large sizes of actinobacteria genomes allowed them to encode a sufficient number of biodegradable enzymes. Phylogenomic analysis of 327 *Rhodococcus* genomes revealed that the *Rhodococcus* genus possessed a small “hard” core genome consisting of 381 orthologous groups (OGs), while a “soft” core genome of 1253 OGs was reached with 99.16% of the genomes [35]. The presence in the *Rhodococcus* genome of a large number of genes encoding degradation enzymes made it possible to consider rhodococci as promising agents in biotechnologies. The genome of the well-studied *Rhodococcus jostii* RHA1 strain contains a remarkably large number of genes coding for oxidative enzymes [36]. This genetic potential allowed this strain to decompose a huge number of organic compounds. Similarly, *Rhodococcus erythropolis* strain MI2 was capable of degrading aromatic and aliphatic compounds, including xenobiotic organic disulphide 4,4′-dithiodibutyric [37]. Strain *R. erythropolis* DCL14 has been found to be able to metabolize alkanes, terpenes, alcohols, and aromatic compounds at high concentrations as sole carbon and energy sources [38]. This diversity of genes made this bacterium attractive for bioremediation processes.

## 4. Conclusions

A few conclusions can be derived from the obtained results. First, the hereditary material, including plasmids of actinobacteria, can be preserved in the cell for a long time even without the impact of external factors, which are believed to be the main driving force for maintaining plasmids in the bacterial cell. Second, since several genes encoding isofunctional enzymes can be present simultaneously in the cell genome, the expression of one/several genes determines the overall substrate specificity of the simultaneously induced enzymatical pool. This allows the cell to adapt to available substrates and use them as a growth source. Third, actinobacteria belong to the so-called microflora of dispersion, which is characterized by the ability to absorb organic substances present in low concentrations. Actinobacteria are characterized by a large set of peripheral metabolism enzymes that provide cells with variability in their degradative reactions. The presence in the soil of actinobacteria of various species containing enzymes differing in substrate specificity in their genomes generally allows bacteria of this group to utilize compounds that differ in structure/character and number of substituents. A wide range of biodegradation enzymes gives these bacteria advantages for surviving in conditions where the concentration of substances is greatly reduced. In sum, the noted features allow actinobacteria to successfully compete with bacteria of other groups for available substrates while at the same time removing pollutants from the environment. The next stage of our work is to conduct laboratory experiments on the purification of soils contaminated with a certain pollutant using the metabolic potential of actinobacteria to assess changes in soil characteristics, primarily its toxicity in tests on standard objects and to evaluate the dynamics of the microbial population of actinobacteria used as a biological product for cleaning polluted soil.

## Figures and Tables

**Figure 1 microorganisms-11-00141-f001:**
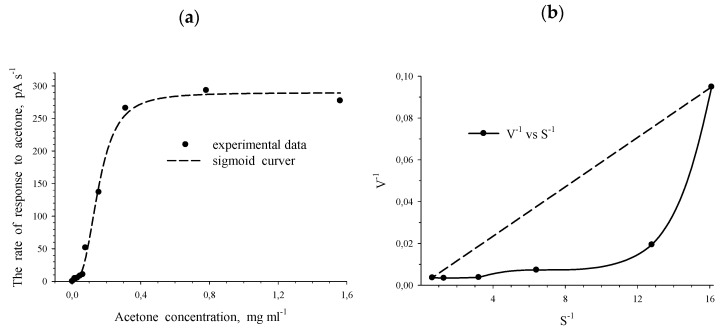
Dependences of the rate of the response (V, pA/c) to acetone on the initial concentration of acetone (S, mg/mL) for acetone-induced *R. opacus* 1CP cells. (**a**) V vs. S dependence; (**b**) V^−1^ vs. S^−1^ dependence.

**Figure 2 microorganisms-11-00141-f002:**
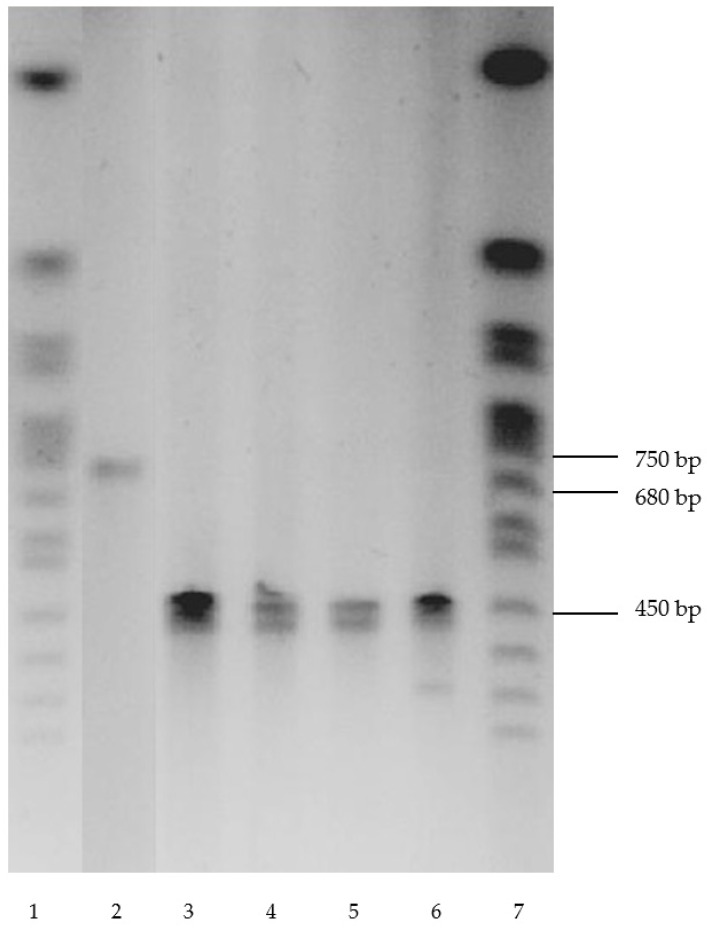
Electropherogram of plasmid DNA: 1, 7—DNA Size Markers—Yeast Chromosomal molecular weight marker (Bio-Rad Laboratories, Hercules, CA, USA), 2—1CP, 3—G10, 4—P1-1, 5—P12, 6—P13.

**Figure 3 microorganisms-11-00141-f003:**
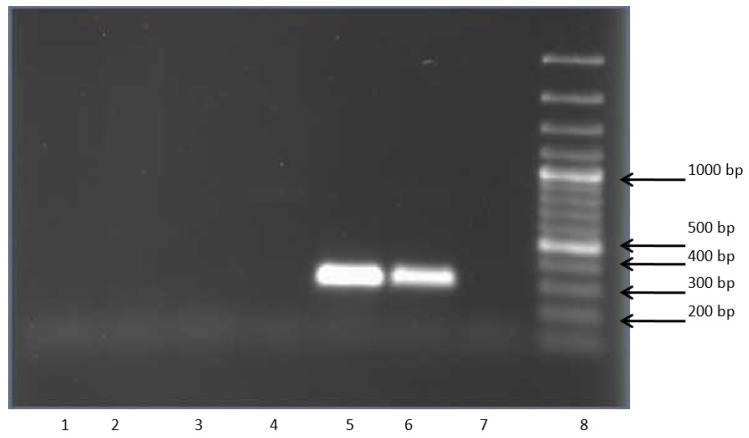
Electropherogram of *clcF* gene amplification products. 1—*Rhodococcus rhodochrous* 89, 2—*Gordonia polyisoprenivorans* 135, 3—*Microbacterium foliorum* B51, 4—*Arthrobacter agilis* Lush13, 5—*Rhodococcus wratislaviensis* G10, 6—*Rhodococcus opacus* 1CP (resting phase), 7—negative control, 8—molecular weight marker 100 b.p. Plus DNA Ladder (Fermentas, Latvia).

**Figure 4 microorganisms-11-00141-f004:**
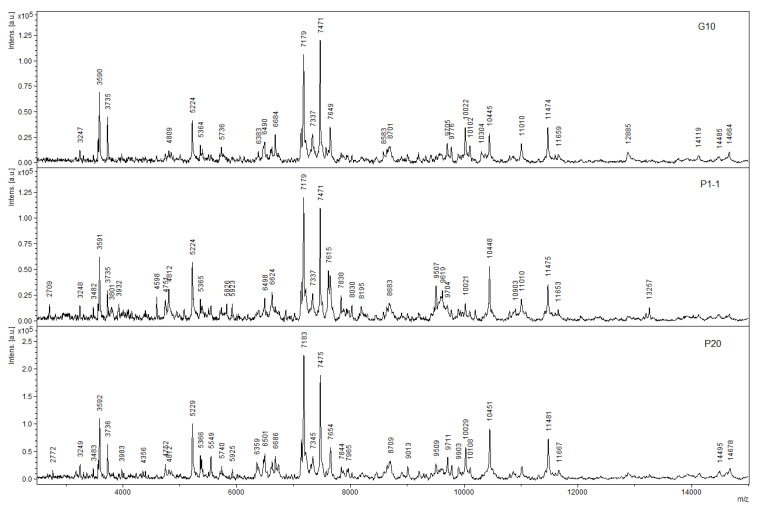
MALDI-ToF spectra of partially purified preparations of ClcF *R. wratislaviensis* G10, P1, P20 (from **top** to **bottom**).

**Table 1 microorganisms-11-00141-t001:** Substrate specificity of 3CBDO of actinobacterial cultures grown in the presence of 3CBA, %.

Substrate	*R. opacus* 1CP	*R. opacus* 6a [15]	*R. rhodochrous* 89	*G. polyisopr*. 135	*R. wratislaviensis* G10
Benzoate	30	24	214	80	92
2-Chlorobenzoate	102	0	0	0	431
3-Chlorobenzoate	100	100	100	100	100
4-Chlorobenzoate	57	0	20	0	0
2,4-Dichlorobenzoate	0	47	n.d.	53	131
2,5-Dichlorobenzoate	0	0	n.d.	0	0
2,6-Dichlorobenzoate	124	n.d.	n.d.	40	469
3,5-Dichlorobenzoate	0	n.d.	n.d.	0	0
2-Hydroxybenzoate	n.d.	48	n.d.	0	n.d.
4-Hydroxybenzoate	94	0	n.d.	0	162
3,4-Dihydroxybenzoate	41	n.d.	n.d.	93	46
2,5-Dihydroxybenzoate	80	n.d.	n.d.	n.d.	85
Phenol	15	n.d.	8	93	0
2Chlorophenol	54	n.d.	10	120	0
3Chlorophenol	61	n.d.	4	133	39
4Chlorophenol	36	n.d.	n.d.	120	200

3-Chlorobenzoate 1,2-dioxygenase (3CBDO) activity was determined by the change in oxygen uptake by cells in response to the addition of a substrate, 3-chlorobenzoate, or substrate analogs. The change in oxygen uptake in response to the addition of 3-CBA was taken as 100%. n.d.: not determined.

**Table 2 microorganisms-11-00141-t002:** Specific activities (Unit/mg of protein) of enzymes in cell-free extracts of strains grown in the presence of 3-CBA.

Enzyme Substrate	Strain			
	*R. opacus* 6a	*R.opacus* 1CP	*R.ruber* P25	*M. foliorum* B51
**CCat 1,2-DO**				
Catechol	0.206 ± 0.008	0.007 ± 0.001	0.005 ± 0.0002	0.20 ± 0.005
**MCI**				
Muconate	0	0.003 ± 0.0001	0	0
2-Methylmuconate	0	0	0.0003 ± 0.00002	0
3- Methylmuconate	0.0042 ± 0.0002	0.043 ± 0.001	0.0003 ± 0.00001	0
2-Chloromuconate	0	0.013 ± 0.001	0.0071 ± 0.0003	0.0221 ± 0.0008
**DLH**				
cis-Dienelactone	0.124 ± 0.005	0.054 ± 0.002	0.0033 ± 0.0001	0.095 ± 0.004

**Table 3 microorganisms-11-00141-t003:** Substrate specificity (in %) of (C) Cat 1,2-DO in cell-free extracts of strains grown with 3-CBA.

Substrate	Strain			
	*R. opacus* 6a	*R.opacus* 1cp	*R. ruber* P25	*M. foliorum* B51
Catechol	100	100	100	100
3-Methylcatechol	100	120	125	131
4-Methylcatechol	n.d.	197	162	230
4-Chlorocatechol	25	52	56	73
3-Chlorocatechol	100	14	n.d.	n.d.
3,5-Dichlorocatechol	4	9	n.d.	n.d.

n.d.: not determined.

**Table 4 microorganisms-11-00141-t004:** Related activity (%) of CCat 1,2-DO of 3CBA grown actinobacteria.

Substrate	Activity after Q-Sepharose Chromatography			
	*R. opacus* 1CP	*R. ruber* P25	*M. foliorum* B 51	
			*Peak 1*	*Peak 2*
Catechol	100	100	100	100
3-Chlorocatechol	94	n.d.	n.d.	n.d.
4-Chlorocatechol	55	87	20	77
3-Methylcatechol	78	225	98	115
4-Methylcatechol	188	262	124	203
3,5-Dichlorocatechol	233	n.d.	3.8	9.8
3,6-Dichlorocatechol	n.d.	50	n.d.	n.d.

## Data Availability

The data presented in this study are available on request from the corresponding author.

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
