# Peer review of "The Contribution of Actinobacteria to the Degradation of Chlorinated Compounds: Variations in the Activity of Key Degradation Enzymes"

_microorganisms, 2023, doi:10.3390/microorganisms11010141_

Round 1

Reviewer 1 Report

This paper, entitled The contribution of actinobacteria to the degradation of chlo-rinated compounds: variations in the activity of key degrada-tion enzymes, is a scholarly work and can oncrease knwoledge on this domain. The authors provide an original and interesting study, the content is relevant to Microorganisms. 

I have some general and specific comments:

- The abstract and keywords are meaningful, but maybe there's too many keywords, please check the number allowed for this journal.

- The manuscript is quite well written and well related to existing literature.

- In the introduction section, please introduce better the objectives and the purpose of this work, in this part, there's only two sentences, maybe the authors could introduce better the work described in the following sections with detailed information. For example, what is the main interest to work on such application and to provide solutions for this? Why focusing on such topic? What is the main interest? Why providing solutions dedicated to chlorinated compounds? What are the environmental and economical considerations that could be added and discussed? This last part of the intricudction should be improved in order to improve the whole quality of the manuscript.

- We understand that strains were already isolated previously and already described in previous studies, but maybe the authors could provide information about storage and preparation of these strains (storage, cutivation, sampling,...).

- About Table 1, please provide units of the data provided here (substrate specificity), it seems to be percentage but there's no information in the title or in the Table.  Please provide how is calculated or measured the substrate specificity?  Is there any control for such measurement or calculation? Why some of them are not determined?

- Please provide accuracy or standard deviation for specific activities of enzyme provided in Table 2. Providing results with the following unit: U/mg of protein requires to define the unit U? It is a common unit used for enzymatic activity but authors should provide details or define this.

- Maybe is it possible to provide accuracy or standard deviation of data in Table 1 and Table 3 for substrate specificity, same comment for Table 4.

- About conclusion, what are the next and future works? What are the perspectives? Please discuss about this. What about the applicability of such technical solution? Is there any other experiments scheduled in the future maybe on real soils with real contaminations or assays in situ?

As it, the manuscript is not fully acceptable for publication and requires minor revision, some amendments and additional information. I recommend the following decision: ACCEPT AFTER MINOR REVISION. I encourage the authors to take into account all the comments and requests of amendments when preparing the revised version of their manuscript if editors give the opportunity to revise this work.

Author Response

Dear Editor, Dear Reviewers,

We deeply appreciate the time and effort that both the reviewers and the editor have spent reviewing our manuscript “The contribution of actinobacteria to the degradation of chlorinated compounds:variations in the activity of key degradation enzymes” Manuscript No. microorganisms-2111978 by Elena V. Emelyanova, S.V.Ramanaiah, Nataliya V. Prisyazhnaya, Ekaterina S. Shumkova, Elena G. Plotnikova, Yonghong Wu and Inna P. Solyanikova.

We sincerely thank the carefully reading of our manuscript and the valuable comments and suggestions. Based on those comments and suggestions, we have carefully revised the manuscript and we believe that this revised version is now more clear and complete. Below, please find the list of answers to the reviewer’s questions (grey text) and revisions. We show the changes marked in yellow in the revised the manuscript accordingly with the reviewer’s comments.

Editor and Reviewer comments  

Reviewer #1

This paper, entitled the contribution of actinobacteria to the degradation of chlorinated compounds: variations in the activity of key degradation enzymes, is a scholarly work and can increase knowledge on this domain. The authors provide an original and interesting study, the content is relevant to Microorganisms. I have some general and specific comments:

Q1. The abstract and keywords are meaningful, but maybe there's too many keywords, please check the number allowed for this journal.

  1. Dear reviewer, thank you for your comment and suggestion, which was helpful in improving the clarity of the manuscript. Following the reviewer comment, the authors reduced the keywords. The modifications are highlighted in yellow in the manuscript.

Q2. The manuscript is quite well written and well related to existing literature.

  1. A. Dear reviewer, we thank you for your comment.

Q3. In the introduction section, please introduce better the objectives and the purpose of this work, in this part, there's only two sentences, maybe the authors could introduce better the work described in the following sections with detailed information. For example, what is the main interest to work on such application and to provide solutions for this? Why focusing on such topic? What is the main interest? Why providing solutions dedicated to chlorinated compounds? What are the environmental and economical considerations that could be added and discussed? This last part of the introduction should be improved in order to improve the whole quality of the manuscript.

  1. Dear reviewer, we thank you for your comment and suggestion. We have carefully revised the manuscript in the introduction part according to your comment. The modifications are highlighted in yellow in the revised manuscript.

Q4. We understand that strains were already isolated previously and already described in previous studies, but maybe the authors could provide information about storage and preparation of these strains (storage, cultivation, sampling...)

  1. Thank you for your comment. We collected the strains of Rhodococcus sp.P1, P20, Rhodococcus ruber P25, Rhodococcus sp.G10, and R.opacus 1CP and Rhodococcus rhodochrous 89 from a soil polluted with polychlorobiphenyls (PCB) (Serpukhov, Moscow Region, Russia) and were isolated from an enrichment culture maintained for several months on 2,4-dichlorophenol as the sole source of carbon and energy, the strain maintained on Luria broth (LB) for more than 20 years after isolation, its cells were used to produce a dormant culture.

 Q5. About Table 1, please provide units of the data provided here (substrate specificity), it seems to be percentage but there's no information in the title or in the Table.  Please provide how is calculated or measured the substrate specificity?  Is there any control for such measurement or calculation? Why some of them are not determined?

  1. A. Required information is added, also in Material and method section. There is no control for this measurement, dust comparison. Unfortunately, we do not have the results for all strains with the same substrates, as the most interest was about mono-chlorinated benzoates and chlorosubstituted phenols. But we decided to include all data to fill the Table as maximum as possible.

Q6. Please provide accuracy or standard deviation for specific activities of enzyme provided in Table 2. Providing results with the following unit: U/mg of protein requires to define the unit U? It is a common unit used for enzymatic activity but authors should provide details or define this.

  1. A. Yes, it is a common unit and it is defined in appropriate section of Materials and methods (“A unit of enzyme activity was defined as the amount of enzyme catalyzing the conversion of 1 µmol of substrate or the formation of 1 µmol of product per minute. Relative activity was calculated as 100% activity with unsubstituted or better substrate.”). The modifications are highlighted in yellow in the revised manuscript.

Q7. Maybe is it possible to provide accuracy or standard deviation of data in Table 1 and Table 3 for substrate specificity, same comment for Table 4.

  1. Since the specific activity of enzymes from different strains differs, it seems more illustrative to present the results in the form of relative activity. Standard deviation is not required in this case.

Q8. About conclusion, what are the next and future works? What are the perspectives? Please discuss about this. What about the applicability of such technical solution? Is there any other experiments scheduled in the future maybe on real soils with real contaminations or assays in situ?

  1. The authors are grateful to the referee for their interest in the work. As for the prospects for its continuation, this work was carried out within the framework of a grant from the Russian Science Foundation, the purpose of which is to develop a technology for cleaning soils from the remnants of pollutants. We believe that actinobacteria, due to the versatility of their degradative apparatus, are promising agents for the development of such technology. The corresponding conclusion has been added to the text.

Q9. As it, the manuscript is not fully acceptable for publication and requires minor revision, some amendments and additional information. I recommend the following decision: ACCEPT AFTER MINOR REVISION. I encourage the authors to take into account all the comments and requests of amendments when preparing the revised version of their manuscript if editors give the opportunity to revise this work.

  1. Dear Reviewer, thank you so much for comments, suggestions and accept after minor revision. We have been revised trough out the manuscript as per the reviewer’s comments and suggestions. The modifications are highlighted in yellow in the revised manuscript.

Reviewer 2 Report

General: This manuscript entitled on “The contribution of actinobacteria to the degradation of chlorinated compounds: variations in the activity of key degradation enzymes” provide a sound knowledge about the enzymatic degradation of chlorinated compounds and activity of enzymes. Authors presented a great effort to prepare this manuscript, however, the reviewer has some queries about this paper. However, authors should have inserted the line numbers in the manuscript. The research gaps, necessity and objectives should be more accurately elaborated. I suggest that this manuscript can be acceptable with minor revision. Other comments are as follows;

-      There are no line numbers in the entire manuscript, which makes the reviewers difficult to point comments on it.

-      This manuscript should be checked by native English expert as there are many syntax and grammatical errors.

-      Please reduce the number of keywords.

-      The topics or subtopics should be written consistently (decide either capitalize the first letter of each word or not to capitalize, as found in section 2.1, 2.8).

-      I would like to suggest the authors to write briefly about existence and properties of 3-chlorobenzoate (3CBA) in the introduction section to justify the necessities and objectives of the study.

-      Page 2: I found an abbreviation “3CDO” for the first time in third paragraph of the introduction section (& only once in this manuscript). Authors need to define the abbreviations where those appear for the first time.

-      Page 2: The two sentences ‘3CBA turned out to be one of the compounds that many researchers used as a "model" to study the ability of microorganisms to decompose chloroaromatic compounds and to identify the features of the enzymes involved in this process. Significant progress has been achieved in this area in a relatively short time.’ need citations. Please add the appropriate references separately for both sentences.

-      I would suggest authors to revise and elaborate the objectives of the present study without constraining into one sentence.

-      Section 2.1.: Please write the country name after “OAO Halogen, Perm”, from where the contaminated soil was obtained. Also, how long (approx. time) the soil was contaminated with (halogen)aromatic compounds and what was the contamination process (naturally or artificially)?

-      200 ml ------à 200 mL. ‘mL’ is more appropriate than ‘ml’. Revise it throughout the entire manuscript.

-      Section 2.2: Please define “CMLI”. Section 2.4: define “BSR” and “MSR”. Similar in MALDI-TOF (section 2.9). The readers should be able to understand the definition of analytical methods.

-      Page 5: The protein concentration was determined according to the modified method of Bradford) [28], using bovine serum albumin as a standard. Revise the typo error.

-      Table 1: Authors should revise the name of chemicals as the numbering in benzene ring is mistyped (e.g. 24 DiClBenz and others). If I am not wrong, a comma should be placed to separate the numbers. It would be better if authors write the full name of all chemicals or proper pre-defined abbreviations.

-      As this study discussed about decomposition of chloroaromatic compounds, I would suggest authors to present GC-MS data for monitoring degradation of those compounds along with the intermediates formed during enzymatic degradation. How did the authors measure their concentration to verify the degradation? Please explain.

Author Response

Response to the Editor and Reviewers comments

Dear Editor, Dear Reviewers,

We deeply appreciate the time and effort that both the reviewers and the editor have spent reviewing our manuscript “The contribution of actinobacteria to the degradation of chlorinated compounds:variations in the activity of key degradation enzymes” Manuscript No. microorganisms-2111978 by Elena V. Emelyanova, S.V.Ramanaiah, Nataliya V. Prisyazhnaya, Ekaterina S. Shumkova, Elena G. Plotnikova, Yonghong Wu and Inna P. Solyanikova.

We sincerely thank the carefully reading of our manuscript and the valuable comments and suggestions. Based on those comments and suggestions, we have carefully revised the manuscript and we believe that this revised version is now more clear and complete. Below, please find the list of answers to the reviewer’s questions (grey text) and revisions. We show the changes marked in yellow in the revised the manuscript accordingly with the reviewer’s comments.

Editor and Reviewer comments  

Reviewer #2

This manuscript entitled on “The contribution of actinobacteria to the degradation of chlorinated compounds: variations in the activity of key degradation enzymes” provide a sound knowledge about the enzymatic degradation of chlorinated compounds and activity of enzymes. Authors presented a great effort to prepare this manuscript, however, the reviewer has some queries about this paper. However, authors should have inserted the line numbers in the manuscript. The research gaps, necessity and objectives should be more accurately elaborated. I suggest that this manuscript can be acceptable with minor revision. Other comments are as follows;

Q1. There are no line numbers in the entire manuscript, which makes the reviewers difficult to point comments on it.

  1. Dear Reviewer, thank you for your general comment. We are deeply sorry for the inconvenience caused by not given the exact line numbers in the manuscript. The revised manuscript has tried to give the line numbers for easier readability, but not possible, due to journal persons is already alignment of manuscript in order to journal format.

Q2.This manuscript should be checked by native English expert as there are many syntax and grammatical errors.

  1. Dear Reviewer, thank you for your valuable suggestion. The manuscript was carefully revised English in the manuscript, to enhance the quality and correct some grammer issues. The modifications are highlighted in yellow in the revised manuscript.

Q3. Please reduce the number of keywords.

  1. Thank you for your comment. Following the reviewer comment, the authors reduced the keywords. The modifications are highlighted in yellow in the manuscript.

Q4.The topics or subtopics should be written consistently (decide either capitalize the first letter of each word or not to capitalize, as found in section 2.1, 2.8).

  1. Thank you so much for your suggestion. We wrote the subtopics sections in consistently in the revised manuscript to enhance clarity and readability. The modifications highlighted in yellow in manuscript

Q4. I would like to suggest the authors to write briefly about existence and properties of 3-chlorobenzoate (3CBA) in the introduction section to justify the necessities and objectives of the study.

  1. A. As per the suggestion, we have been carefully revised the introduction section has been improved. The modifications highlighted in yellow in manuscript.

Q5. Page 2: I found an abbreviation “3CDO” for the first time in third paragraph of the introduction section (& only once in this manuscript). Authors need to define the abbreviations where those appear for the first time.

  1. A. Thank you for your suggestions, 3-Chlorobenzoate 1,2-dioxygenase (3CBDO). The full names of the abbreviations as per the journal guidelines has been provided in its first used, please once check in the sections, the modifications highlighted in yellow in manuscript.

Q6. Page 2: The two sentences ‘3CBA turned out to be one of the compounds that many researchers used as a "model" to study the ability of microorganisms to decompose chloroaromatic compounds and to identify the features of the enzymes involved in this process. Significant progress has been achieved in this area in a relatively short time.’ need citations. Please add the appropriate references separately for both sentences.

  1. A. Thank you very much for the reviewer suggestion. We have been carefully revised by adding recent references. The modifications highlighted in yellow in manuscript.

Q7. I would suggest authors to revise and elaborate the objectives of the present study without constraining into one sentence.

  1. A. Dear reviewer, we thank you for your comment and suggestion. We have revised the manuscript in the introduction ending part to elaborate the objectives of the present study according to your comment. The modifications are highlighted in yellow in the revised manuscript.

Q8. Section 2.1.: Please write the country name after “OAO Halogen, Perm”, from where the contaminated soil was obtained. Also, how long (approx. time) the soil was contaminated with (halogen) aromatic compounds and what was the contamination process (naturally or artificially)?

  1. A. Thank you very much for the valuable suggestions. We have been revised the sentence as per the reviewer suggestion to avoid confusion and all the changes we included in the revised manuscript. The modifications are highlighted in yellow in the revised manuscript.

Q9. 200 ml ------à 200 mL. ‘mL’ is more appropriate than ‘ml’. Revise it throughout the entire manuscript.

  1. A. Thank you very much for careful revision. We have been corrected the specified sentence of ml to mL for avoid confusion to the readers as per the reviewers suggestion. The modifications are highlighted in yellow in the revised manuscript.

Q10. Section 2.2: Please define “CMLI”. Section 2.4: define “BSR” and “MSR”. Similar in MALDI-TOF (section 2.9). The readers should be able to understand the definition of analytical methods.

  1. Thank you for your very helpful comments. The full names of the abbrevations as per the reviewer guidelines has been changed in the revised manuscript, please once check in the section 2.2, 2.4 and 2.9. The modifications highlighted in yellow in manuscript.

Q11. Page 5: The protein concentration was determined according to the modified method of Bradford) [28], using bovine serum albumin as a standard. Revise the typo error.

  1. Thank you very much for the reviewer suggestion. We apology for the inconvenience caused by typo errors. As per the suggestion, typo error has been corrected in the revised manuscript. The modifications highlighted in yellow in manuscript.

Q12. Table 1: Authors should revise the name of chemicals as the numbering in benzene ring is mistyped (e.g. 24 DiClBenz and others). If I am not wrong, a comma should be placed to separate the numbers. It would be better if authors write the full name of all chemicals or proper pre-defined abbreviations.

  1. Thank you so much for the valuable suggestions. We apology for the inconvenience caused by typo errors. As per the suggestion, typo error has been corrected in the revised manuscript. The modifications highlighted in yellow in manuscript.

Q13.  As this study discussed about decomposition of chloroaromatic compounds, I would suggest authors to present GC-MS data for monitoring degradation of those compounds along with the intermediates formed during enzymatic degradation. How did the authors measure their concentration to verify the degradation? Please explain.

  1. A. Thank you for your comment. An analysis of the intermediates formed during the decomposition of 3-CBA by rhodococci 1CP and 6a was published earlier. In this work, emphasis was placed on the induction of key enzymes. We attempted to adapt novel method for rapid screening of strains for their ability to degrade target substrates and to determine the pathway by which they are degraded. An analysis for the presence of CMLD was used as a standard test. In this situation, this enzyme could serve as the basis for separating the flows along which decomposition of chloroaromatic compounds in actinobacteria proceeds: via the path of ortho-cleavage of 3-CCat, 4-CCat or with the stage of initial dehalogenation.
